# Phytochemical Analysis and In Vitro Effects of *Allium fistulosum* L. and *Allium sativum* L. Extracts on Human Normal and Tumor Cell Lines: A Comparative Study

**DOI:** 10.3390/molecules26030574

**Published:** 2021-01-22

**Authors:** Adrian Bogdan Țigu, Cristian Silviu Moldovan, Vlad-Alexandru Toma, Anca Daniela Farcaș, Augustin Cătălin Moț, Ancuța Jurj, Eva Fischer-Fodor, Cristina Mircea, Marcel Pârvu

**Affiliations:** 1Faculty of Biology and Geology, Babeș-Bolyai University, 42 Republicii Street, 400015 Cluj-Napoca, Romania; vlad.al.toma@gmail.com (V.-A.T.); farcasanca14@gmail.com (A.D.F.); cristina.mircea@ubbcluj.ro (C.M.); 2Research Center for Advanced Medicine—MedFuture, “Iuliu Hatieganu” University of Medicine and Pharmacy Cluj-Napoca, Louis Pasteur Street 6, 400349 Cluj-Napoca, Romania; moldovan.cristian1994@gmail.com (C.S.M.); fischer.eva@iocn.ro (E.F.-F.); 3National Institute for Research and Development of Isotopic and Molecular Technologies, 400293 Cluj-Napoca, Romania; 4Institute of Biological Research, Branch of NIRDBS Bucharest, 400113 Cluj-Napoca, Romania; 5Centre for Systems Biology, Biodiversity and Bioresurces “3B”, Babeș-Bolyai University, 400000 Cluj-Napoca, Romania; 6Department of Chemistry, Faculty of Chemistry and Chemical Engineering, Babes-Bolyai University, 11 Arany Janos Street, 400028 Cluj-Napoca, Romania; gusty_chem@yahoo.com; 7The Research Center for Functional Genomics, Biomedicine and Translational Medicine, “Iuliu Hatieganu” University of Medicine and Pharmacy, 400012 Cluj-Napoca, Romania; ancajurj15@gmail.com; 8Department of Radiobiology and Tumor Biology, The Oncology Institute “Prof Dr Ion Chiricuta”, 400015 Cluj-Napoca, Romania

**Keywords:** *Allium*, necrosis, apoptosis, toxicity, allicin, antiproliferative

## Abstract

*Allium sativum* L. (garlic bulbs) and *Allium fistulosum* L. (Welsh onion leaves) showed quantitative differences of identified compounds: allicin and alliin (380 µg/mL and 1410 µg/mL in garlic; 20 µg/mL and 145 µg/mL in Welsh onion), and the phenolic compounds (chlorogenic acid, *p*-coumaric acid, ferulic acid, gentisic acid, 4-hydroxybenzoic acid, kaempferol, isoquercitrin, quercitrin, quercetin, and rutin). The chemical composition determined the inhibitory activity of *Allium* extracts in a dose-dependent manner, on human normal cells (BJ-IC_50_ 0.8841% garlic/0.2433% Welsh onion and HaCaT-IC_50_ 1.086% garlic/0.6197% Welsh onion) and tumor cells (DLD-1-IC_50_ 5.482%/2.124%; MDA-MB-231-IC_50_ 6.375%/2.464%; MCF-7-IC_50_ 6.131%/3.353%; and SK-MES-1-IC_50_ 4.651%/5.819%). At high concentrations, the cytotoxic activity of each extract, on normal cells, was confirmed by: the 50% of the growth inhibition concentration (IC_50)_ value, the cell death induced by necrosis, and biochemical determination of LDH, catalase, and Caspase-3. The four tumor cell lines treated with high concentrations (10%, 5%, 2.5%, and 1.25%) of garlic extract showed different sensibility, appreciated on the base of IC_50_ value for the most sensitive cell line (SK-MES-1), and the less sensitive (MDA-MB-231) cell line. The high concentrations of Welsh onion extract (5%, 2.5%, and 1.25%) induced pH changes in the culture medium and SK-MES-1 being the less sensitive cell line.

## 1. Introduction

*Allium* plants have been used in traditional medicine since ancient times. Currently, studies regarding the phytotherapeutical properties of *Allium* species are investigating the effects of *Allium* extracts and bioactive compounds [1,2]. *Allium* species are still studied due to their bioactive compounds [1]. Most of the studies regarding the phytotherapeutical properties of *Allium* species are focused on *Allium sativum* (garlic) and *Allium cepa* (onion). *Allium* plants are rich in bioactive compounds like alliin, allicin, ajoene, sterols, flavonoids, polyphenols, and polycarboxylic acids, which are responsible for the biological effects of different *Allium* species [3].

Garlic is one of the most cultivated therapeutical plants [4]. The studies focused on the curative capacities of garlic highlighted its antioxidative properties [5], hypoglycemic and cardioprotective effects [6,7], anticancer [8], antifibrotic [9], and antifungal properties [10].

The antitumoral effect of garlic induced by its bioactive sulfur compounds like allicin, diallyl-sulfides (DAS), diallyl-disulfides (DADS), and diallyl-trisulfides (DATS), raises interest for many studies [11,12,13]. DATS can stimulate cytotoxicity by targeting reactive oxygen species (ROS) and can activate the ROS dependent caspase pathway, therefore it can promote apoptosis [14]. Sulfur compounds act as upregulators for antioxidant genes, can promote Nrf2-ARE activation, and are potent inhibitors of the inflammatory process [15,16]. Not only the sulfur compounds but also the flavonoids and saponins present in *Allium* species were investigated for their cytotoxic, antiproliferative, proapoptotic, antiangiogenic, or antimetastatic potential on numerous human cancers line [2,17]. Usually, the anticancer mechanisms activate the apoptosis-related genes caspases (caspase-3, -8, and -9) and bcl-2 [18].

Isolated bioactive compounds present in garlic, and obtained extracts, were investigated for their anticancer potential. Allicin is the result of hydrolyzed alliin under alliinase enzyme action, with ajoene as a secondary compound resulted after alliin decomposition [15,19]. Allicin has an immunostimulatory effect [19] and antitumoral properties by inhibiting cell proliferation, migration, and enhancing apoptosis [12,20,21,22]. The antitumor effect of allicin was investigated on different tumor cell lines like cervical cancer, breast cancer, colorectal cancer, gastric cancer, glioblastoma, leukemia, lymphoma, and endometrial adenocarcinoma [23,24,25,26,27,28,29,30,31,32,33].

Various *Allium* extracts, aqueous or alcoholic, were tested individually as a single agent treatment or combined with cancer therapeutically drugs in several cancer lines like oral and lung [34], breast, cervical, colon and liver [35,36,37,38,39], leukemia [40], and squamous carcinoma [41]. *Allium sativum* extracts were also proven to have cytotoxic effects on multidrug-resistant human cancer cells by altering the mitochondrial permeability [42]. 

Welsh onion (*Allium fistulosum* L.) is less studied compared to garlic. It is a common *Allium* plant in Eastern Europe, with antifungal and antimicrobial properties, due to its high concentration of sterols and sulfuric compounds [43]. The studies investigating *A. fistulosum* biological properties highlighted mostly the anti-inflammatory and antioxidant effects, while the antitumor potential was elevated on the MDA-MB-453 metastatic breast cancer cell line, in which the Welsh onion stimulated early apoptosis [44].

Natural compounds represent potential candidates in fighting against various diseases. A pure compound, isolated and well characterized, can serve as a therapeutic agent alone or in combination with conventional therapy, due to its biological properties [45,46,47,48,49,50,51].

The aim of this study is to investigate the in vitro effect of two *Allium* species that present high interest as being therapeutical agents. The presented characterization and investigation of *A. sativum* and *A. fistulosum* can provide scientific information that may lead to further studies.

## 2. Results and Discussions

### 2.1. Phytochemical Characterization

Both *Allium* extracts were analyzed to investigate the concentration of alliin and allicin. Four samples of each extract were prepared according to the protocol and the concentration of compounds was expressed in µg/mL (Table 1).

Garlic extract contained 1410 µg/mL of alliin and 380 µg/mL of allicin, a lot more than *A. fistulosum* extract that contained only 145 µg/mL of alliin and 20 µg/mL of allicin (Table 1).

Same as for alliin and allicin determinations, four samples of both extracts were prepared and analyzed for phenolic acids and flavonoids content. As presented in Table 1, *A. fistulosum* is rich in isoquercitrin (280 µg µg/mL), ferulic acid (230 µg/mL), and rutin (215 µg/mL), whereas *A. sativum* has 65 µg/mL of chlorogenic acid, 44 µg/mL of *p*-coumaric acid, and 25 µg/mL of 4-hydroxybenzoic acid.

### 2.2. Evaluation of the Acute Effect on Human Normal Cell Viability

To explore the *Allium* extracts toxicity on human normal fibroblasts (BJ) and human normal keratinocytes (HaCaT), the viability of these cells was assessed by an MTT assay at 24 h. As presented in Figure 1, the *A. sativum* extract inhibited the growth of BJ and HaCaT cells in a dose dependent manner, with the 50% of the growth inhibition concentration (IC_50_) value of 0.8841% in the growth medium for BJ cells, equivalent to 8.841 mg/mL and 1.086% in the growth medium for HaCaT cells at 1.086% equivalent to 10.86 mg/mL. When calculating the equivalent of extract in mg/mL we used the concentrations of stock solutions for each extract, 1 g garlic bulbs in each mL and 1 g of Welsh onion leaves in each 1.2 mL.

*A. fistulosum* extract also presents inhibitory activity on both cell lines included in this study (Figure 2). IC_50_ value for BJ was 0.2433% in the growth medium; equivalent to 2.019 mg/mL, while for HaCaT cells IC_50_ value was 0.6197% equivalent to 5.144 mg/mL.

Both *A. sativum* and *A. fistulosum* extracts have dose-dependent antiproliferative potential. At concentrations higher than 1.25% extract in the cell culture medium, garlic extract had lower inhibitory potential on BJ and HaCaT proliferation. On the other hand, Welsh onion extract fully inhibited cell growth at 5% and 10% of extract in the cell culture medium, while below 1.25% of extract, the growth inhibitor was lowered in a dose dependent manner. The difference between extracts was visible at their IC_50_ values.

Normal human fibroblasts (BJ cells) and keratinocytes (HaCaT) were sensitive to garlic and Welsh onion extract with different half inhibitory doses (IC_50_). Viability tests that were used in our study highlight the dose-dependent viability rate, with a very increased cell death for 100 mg/mL garlic extract and 86 mg/mL Welsh onion extract, which represent the 10% dose added to cells.

Growth inhibition was observed on the confocal microscopy analysis (Figure 3 and Figure 4), morphological changes highlighting induced cell death in a dose dependent manner.

### 2.3. Morphological Evaluation

When cells are exposed to cytotoxic agents they tend to adapt if this is possible. At high concentrations of toxic compounds apoptosis is induced. During cell death the cytoskeleton collapses and the nuclear envelope disassembles and the DNA breaks into pieces, moreover the cell surface is modified. The intracellular mechanism of apoptosis is similar to almost all cells, caspases cleave key proteins in the cell, and some caspases cleave nuclear lamina or proteins that in normal situations keep DNAse in an inactive form, saving the DNA [52].

The most cytotoxic concentration seems to be the 10% garlic extract where the cytoskeleton and nucleus are almost gone, and the mitochondrial network is no more highlighted (D for BJ and H for HaCaT). The IC_50_ concentration for both cell lines had no cytotoxic effect; the nucleus and cytoskeleton are very well highlighted, and the mitochondrial network seems to be very active (C for BJ and G for HaCaT). Cell structure and the morphological characteristics for cells that were treated with a lower extract dose than IC_50_ (B for BJ and F for HaCaT) were comparable with the control group (A for BJ and E for HaCaT). These images confirmed the cytotoxic effect for both extracts at high concentration (10% extract) and less cytotoxic at IC_50_ concentrations (0.8841% for BJ and 1.086% for HaCaT) and below IC_50_ (0.5% for BJ and 0.1% for HaCaT), morphological modification with the disrupted cytoskeleton and mitochondrial network was indicating that cell death was induced by necrosis at 10% of extract. Morphological changes were less intense at IC_50_ concentrations and below compared to 10% of extract.

The cell integrity was investigated with a triple staining protocol, focusing on the mitochondrial networks, the cytoskeleton and the nucleus [12,51]. *A. sativum* is widely used for its antibacterial, antifungal, anti-inflammatory, and antitumoral properties, especially due to its organosulfur compounds [4,10,20,45]. Garlic extract showed a growth inhibitory effect after 24 h exposure to 100 mg/mL (equivalent to 10% extract), cell death aspects were visible, as presented in Figure 3D,H, the cytoskeleton collapsed, on BJ cells the nucleus was slightly visible, moreover the mitochondrial networks were not visible and the cell density was significantly reduced compared to the control group (Figure 3A). At the IC_50_ concentration (0.8841% for BJ and 1.086% for HaCaT) for both BJ and HaCaT cells the cytoskeleton networks showed minor damage and the cellular shape was modified (Figure 3C,G). At concentrations below IC_50_ (0.5% for BJ and 0.1% for HaCaT) the cytoskeleton was not damaged, while the mitochondrial networks were visible and the morphological characteristics (Figure 3B,F) were comparable with the control group. HaCaT cells showed particular changes in mitochondrial networks at 5 mg/mL treatment with an intensified activity compared to the control group.

As presented in Figure 3, *A. sativum* extract inhibited cell growth in a dose dependent manner, this effect could be attributed to the complex mixture of bioactive compounds and due to the hydroalcoholic composition of the extract, thus inducing cellular stress.

*A. fistulosum* is a common *Allium* plant in Eastern Europe, rich in sulfur compounds and sterols, with antibacterial and antifungal properties [43]. As presented in Figure 4, at the highest dose of Welsh onion extract (83 mg/mL) induced cell death in both cell lines, the cytoskeleton was completely damaged, the mitochondrial networks and the nucleus were unclear and damaged (Figure 4D,H). At the IC_50_ dose, 2.019 mg/mL for BJ and 5.144 mg/mL for HaCaT, the cytoskeleton showed moderate damage, with focal adhesion points at HaCaT cells, which might be visible due to the usual distribution of these cells in small groups, with strong connections between cells (Figure 4C,G). When adding 1 mg/mL of the extract on both cell lines (Figure 4B,F), the cytoskeleton showed minor damage, the mitochondrial networks were visible, and the cell distribution and shape were comparable with the control group.

Welsh onion extract had a cytotoxic effect in a dose dependent manner, with small differences between the selected cell lines, BJ cells being more affected by the treatment compared with HaCaT; moreover, HaCaT cells had active mitochondrial networks highlighted at 1 mg/mL dose, while BJ cells had minor cytoskeleton damage and few mitochondrial networks with higher intensity (Figure 4).

The most cytotoxic concentration seems to be the 10% Welsh onion extract where the cytoskeleton was disrupted and the nucleus was almost gone and the mitochondrial network was not as usual but was still visible for BJ cells and for HaCaT cells no mitochondria network could be highlighted (D for BJ and H for HaCaT). The IC_50_ concentration for both cell lines had no cytotoxic effect; the nucleus and cytoskeleton were very well highlighted, and the mitochondrial network seemed to be very active (C for BJ and G for HaCaT). Cell structure and the morphological characteristics for cells that were treated with a lower extract dose than IC_50_ (B for BJ and F for HaCaT) were comparable with the control group (A for BJ and E for HaCaT).

The highest concentrations that were used (10%, 5%, and 2.5% extract) were the most cytotoxic, cytoskeleton modifications were induced, and the cell shape was different compared to the control. Moreover, mitochondrial networks were less intense than control and the nucleus for treated cells with 100 mg/mL garlic extract and 86 mg/mL Welsh onion extract was no more visible or slightly visible. Confocal images are leading us to say that *Allium* extracts at a higher dose than IC_50_ were inducing necrosis in normal fibroblasts and keratinocytes.

These images were indicating a cytotoxic effect for both extracts when concentration was high (10% extract) and a lower cytotoxic effect when the extract concentration was reduced, morphological modification with the disrupted cytoskeleton and mitochondrial network was indicating that cell death was induced by necrosis in the group treated with 10% extracts, while at lower concentrations of extracts cell populations were less affected and the structural damage was highlighted in a dose-dependent manner.

### 2.4. Biochemical Determination of LDH, CAT, and Caspase 3

Caspase 3 is a key molecule in the apoptotic pathway, being the node between the intrinsic and extrinsic apoptotic pathway. Increased activity of active Caspase 3 (Casp3) is correlated with apoptotic cells [53]. *Allium* species can induce apoptosis due to their organosulfur compounds, which increase Bax/Bcl-2 ration and enhance Casp3 activity [54].

*A. sativum* and *A. fistulosum* extracts slightly increased the activity of Casp3 in both cell lines, with a pronounced effect on keratinocytes when all three doses had increased the Casp3 activity compared to the control (Figure 5). In order to confirm the apoptosis or the necrosis, the flow cytometry analysis should be taken into consideration, moreover, these results need to be correlated with lactate dehydrogenase (LDH) activity and catalase (CAT) activity for a complete understanding of the effect of both extracts on BJ and HaCaT cells.

To discriminate the necrosis and apoptosis, LDH and CAT activity were measured by colorimetric assay. Catalase is an important enzyme involved in cell protection against oxidative damage. CAT catalyzes the decomposition of H_2_O_2_ into H_2_O and O_2_ [55]. Increased CAT activity is an indicator of increased oxidative stress, which may dysregulate normal physiological processes within the cells. An overexpressed CAT can reveal a high concentration of reactive oxygen species (ROS). Garlic extract has no effect on CAT activity in BJ cells at 10% of extract in the cell culture medium, while at IC_50_ concentration the CAT activity was increased in BJ cells. On the other hand, on HaCaT cells, at 10% of garlic extract, CAT activity was increased, and decreased at lower doses (Figure 6). In the case of Welsh onion extract, the effect was similar for both cell lines, at the highest concentration the CAT activity was increased, while at lower doses a decrease of CAT activity was observed.

At the highest dose, extracts acted like pro-oxidants and increased CAT activity, while around the IC_50_ and below the CAT activity was significantly lowered compared to the control group. These indicate that in a dose dependent manner, *Allium* extracts act differently depending on the cells that are used, moreover the shift between pro-oxidant and antioxidant effect can be made by changing the concentration of the extract that is used in the experiment. Increased CAT is correlated with high oxidative stress in cells. Lactate dehydrogenase (LDH) is a soluble enzyme located in the cytoplasm in almost all cells and it is released in the extracellular space when the plasma membrane is disrupted. Damaged membranes could be a marker for necrosis, moreover, during necrosis, the function and morphology of the organelles suffer changes, some of them have disrupted membranes or modified shape [56].

LDH release can be used as a marker for cytotoxicity; very active LDH in the supernatant is indicating that cell membranes have suffered modifications and cellular integrity was affected [41], both *Allium* extracts that were used in this study have a cytotoxic effect depending on the dose that was added as treatment, with a more pronounced cytotoxic effect for the Welsh onion extract. Furthermore, HaCaT cells seem to be more sensitive to the *A. fistulosum* extract than BJ cells, as presented in Figure 7.

Cells that were treated with 100 mg/mL garlic and 86 mg/mL Welsh onion had released a large amount of LDH in the culture medium and had CAT with an increased activity indicating that oxidative stress was harming the cells and was disrupting cell membrane causing cell death. Caspase 3 was not so intense in its activity in none of the cells. LDH and CAT that were increased were indicating that cells might be in necrosis. To confirm or not confirm the necrosis, Annexin V/PI was performed, or samples were treated with the highest dose of 50 mg/mL garlic extract and 43 mg/mL Welsh onion extract. The results were encouraging us to confirm that *Allium* extracts had cytotoxic effects by inducing necrosis in treated cells in a dose-dependent manner and with a different intensity in BJ cells compared to HaCaT cells.

Doses between 0.062 and 1 mg/mL were used to investigate the ROS production after exposing the SCC-15 cell line to *A. sativum* extract, with results indicating a dose-dependent increasing ROS production, correlated with an increased LDH activity [41], similar to herein presented results on the LDH activity trend.

### 2.5. The Evaluation of Induced Cell Death

For completing the confocal images information and to characterize which type of cell death was induced in BJ and HaCaT cell lines, biochemical analysis, Caspase 3 quantification, and Annexin/PI Flow Cytometry Assay are coming to underline the effect of *A. sativum* and *A. fistulosum* extracts.

Annexin V/PI Assay is a golden standard assay to determine the percentage of viable, necrotic, or apoptotic cells. Untreated cells from HaCaT negative control group were 95% viable with 5% necrotic cells, while the positive control for HaCaT cells had 88.6% viable cells, 10.8% necrotic, 0.4% early apoptotic, and 0.2% late apoptotic cells. Necrosis was induced in a dose dependent manner in HaCaT cells by the *A. sativum* extract with the most intense necrosis for the 5% concentration (6.1% necrotic cells and 93.9% viable) and by the *A. fistulosum* extract with 7.9% necrotic cells at the sample treated with 5% extract. It is interesting that for HaCaT cells no apoptotic cell was detected, while for BJ cells only for the 0.5% garlic extract no apoptotic cell was detected. The negative control for BJ cells had 90.2% viable cells, 9.7% necrotic cells, and 0.1% late apoptotic cells, while the positive control had 84% viable and 16% necrotic cells. The most aggressive treatment was the 5% *A. sativum* extract with 19.3% necrotic cells, 0.1% early apoptotic, and 0.4% late apoptotic cells; moreover the *A. fistulosum* extract at 5% concentration in cell culture media induced necrosis in 16.8% of the cells and 0.5% late apoptosis. It is true that at the IC_50_ concentration, both garlic (1% extract concentration) and Welsh onion (0.25% extract concentration) induced early apoptosis, 1.6% for BJ treated with garlic at IC_50_ concentration and 2.6% early apoptosis and 1% late apoptosis for BJ treated with the Welsh onion extract at IC_50_ concentration. However, concentrations below IC_50_ had a less toxic effect, moreover, all concentrations that were used kept cell viability above 80%.

Garlic and Welsh onion extracts induced necrosis, according to the Annexin V/PI flow cytometry assay, with a very low rate of induced apoptosis. In Figure 8 it could be observed very clearly how both extracts reduced cell viability (dark green plots) and how intense is the necrosis (red plots), moreover the light blue plots indicate early apoptosis and light green plots late apoptosis, showing the massive difference between necrotic and apoptotic cell number, moreover, it indicates that the cell death morphological aspects and the elevated LDH activity at high concentrations of extracts were related to cell death by necrosis.

### 2.6. Antiproliferative Effect on Tumor Cells

Different concentrations of *A. sativum* and *A. fistulosum* extracts were investigated for their potential antiproliferative effect on four tumor cell lines, colorectal cancer cells (DLD-1), breast cancer cells (MDA-MB-231 and MCF-7), and lung carcinoma cells (SK-MES-1). For garlic hydroalcoholic extract, concentrations between 10% and 0.156% extract in cell culture medium were analyzed (Figure 9), while for Welsh onion extract the concentrations were between 2.5% and 0.156% (Figure 10), when adding 5% and 10% extract in the cell culture medium, we observed color change due to the pH indicator. We excluded those values from our analysis.

The IC_50_ for both *Allium* extracts were higher than 1.25% extract (equivalent to more than 1 mg/mL). *A. sativum* extract had IC_50_ values of 4.651% for SK-MES-1, which was the most sensitive cell line to garlic extract, 5.482% for DLD-1, 6.131% for MCF-7, and 6.375% for MDA-MB-231, which was the less sensitive cell line to garlic extract. Welsh onion extract had IC_50_ values of 2.124% for DLD-1 which is the most sensitive cell line to Welsh onion extract, 2.464% for MDA-MB-231, 3.353% for MCF-7, and 5.819% for SK-MES-1, which was the less sensitive cell line to Welsh onion. This value indicates that the extracts were concentrated in bioactive compounds, which might inhibit tumor cells, furthermore the extracts had a dose-dependent inhibitory effect on tumor cells. Alliin and allicin were the main compounds in both *Allium* extracts, the inhibitory effects were different and the inhibition rate was proportional with the concentration of extract added to the cells. The bioactive sulfuric compounds such as allicin and allyl sulfides have an antitumoral effect and are responsible for the growth inhibition of the studied tumor cells [4,57,58,59].

*Allium* extract used as a treatment, at a certain concentration can become cytotoxic for cells. At lower doses, no toxicity was observed on normal human cells. *Allium* extracts, particularly garlic, were investigated for their biological properties such as antioxidative [60,61], anti-inflammatory [62], antitumoral [63], antibacterial [64], and other biological properties due to their chemical composition.

The antiproliferative effect of *A. sativum* and *A. fistulosum* extracts was focused on active compounds within their composition and was highlighted in a dose-dependent manner.

It was demonstrated that both *Allium* extracts have an immunostimulatory effect [65] mostly due to their sulfur bioactive compounds, like allicin [19], being a possible therapeutical approach for alternative antitumor therapy. Further investigations and optimization of the therapeutic dose should be taken into consideration, moreover, a multidose experiment could present interest in investigating the antitumoral effect of *Allium* extracts and comparing the effects with their bioactive compounds and conventional therapeutical agents.

Allicin has the potential to play a role in antitumor therapies as lung, breast, and colorectal carcinoma [12,20]. Our previously published results indicate that the bioactive sulfuric compound can inhibit tumor cell proliferation, colony formation, and migration, moreover it can induce cell death by triggering apoptotic and non-apoptotic pathways [12]. In the last years, many researchers have demonstrated the in vitro antitumoral potential of allicin, which can inhibit human renal cell carcinoma via the HIF pathway [66], can inhibit thyroid cancer progression [67], can suppress migration and invasion in cervical cancer cells [68], or can sensitize hepatocellular cancer cells to the antitumor activity of cytostatic drugs like 5-fluorouracil [69].

*A. sativum* extract had 380 μg/mL allicin, while *A. fistulosum* had 20 μg/mL allicin. The antiproliferative effect of both extracts might be attributed to allicin, which was the main bioactive compound in *Allium* extracts. At the IC_50_ values, allicin was present at a concentration of 19 μg/mL equivalent for 5% *A. sativum* extract and at concentrations between 0.2 and 1 μg/mL equivalent for 2–5% *A. fistulosum* extract.

The antiproliferative effect of both *Allium* extracts was visible in a dose-dependent manner, depending on the concentration of the main bioactive compound, allicin. Other studies revealed that allicin has an antiproliferative effect against tumor cells at different concentrations, depending on the cell type. Allicin showed an inhibitory effect against U251 cells at an IC_50_ of 41.97 μg/mL [70]; 10.38 μg/mL against SK-Hep-1 cells; and 10.00 μg/mL against BEL-7402 cells [69]. In our previous study, allicin inhibited SK-MES-1 cells at an IC_50_ of 1.39 μg/mL (equivalent to 8.6 μM) and DLD-1 cells at IC_50_ of 8.67 μg/mL (equivalent to 53.53 μM) [12]. Besides, allicin showed a synergistic effect when combined with 5-fluorouracil on DLD-1 and SK-MES-1 [12].

The antiproliferative effect of allicin from *Allium* extracts is completed by the phenolic compounds [2,18]. Moreover, glucoside quercetin has an important role in the antiproliferative effect of *A. sativum*, *A. fistulosum*, *A. cepa*, and *A. chinense* extracts against HepG2, PC-3, and HT-29 cell lines [69]. The dose-dependent inhibitory effect of *A. sativum* and *A. fistulosum* extracts is related to the concentration of allicin. Gruhlke et al. evaluated the inhibitory effect of allicin in garlic juice and synthesized allicin on lung adenocarcinoma (A549 cell line), human mammary carcinoma (MCF-7 cell line), human colorectal carcinoma (HT29 cell line), mouse fibroblasts (NIH 3T3 cell line), and human umbilical vein endothelial cells (HUVEC) and the results indicate an inhibitory rate on all cell lines, in a dose-dependent manner [20]. Furthermore, *Allium* extracts are rich in polyphenols, which play important role in the biological effects of garlic and Welsh onion. The antitumor effect of both *Allium* extracts is mainly attributed to allicin, while polyphenols are enhancing this biological effect. Fujimoto et al. demonstrated that polyphenols have a cytotoxic effect in a dose-dependent manner both on normal and tumor cells, and at a well-established concentration, polyphenols could inhibit tumor cell without affecting normal cells [71].

Polyphenols, in a dose-dependent manner, can induce cell death by inducing apoptosis [71]. Sulfuric compounds, like allicin, have a different effect on cell signaling pathways. Allicin inhibits cell growth and induces apoptosis via the ERK-dependent pathway [31] or can activate both intrinsic and extrinsic apoptotic pathways [25,33,70]. Moreover, allicin can inhibit tumor cell progression by suppressing the HIF pathway [66]. Allicin suppresses cervical cancer cells invasion and migration via inhibiting NRF_2_ [68]. Furthermore, allicin has an immunostimulatory effect via Colec12, MARCO, and SCARB1 receptors [19]. A similar proapoptotic effect was observed in the case of diallyl trisulfides, which are part of *Allium* plants’ chemical composition, Shin et al. demonstrated that diallyl trisulfides induce apoptosis in caspase-dependent manner and create cross-talks with PI3K/Akt and JNK pathways [58].

Both *Allium* extracts that were investigated in our study were rich in allicin and polyphenols and showed an inhibitory effect in a dose-dependent manner both on normal and tumor cells. At high concentrations, above the concentration obtained for the IC_50_, cell damage was induced and the LDH and CAT activity was increased. On the other hand, at lower concentrations the cellular stress was reduced, LDH and CAT activity were significantly decreased compared to the 10% extract. The inhibitory effect was related to the chemical composition of the extracts. The cytotoxicity could be reduced by choosing a therapeutic dose that is not toxic to normal cells and by isolating or synthesizing bioactive compounds from the extracts, compounds that are responsible for the biological effect of these plants.

Further investigations are needed to evaluate the therapeutical potential of *Allium* extracts and their bioactive compounds, focusing on the molecular links between the active compounds and the effector molecules in eukaryotic cells.

## 3. Materials and Methods

### 3.1. Plant Material and Growth Conditions

Garlic (*Allium sativum* L.) plants were cultivated in a private garden [10] and Welsh onion (*Allium fistulosum* L.) plants in the Botanical Garden “Alexandru Borza” from luj-Napoca, Romania. For each *Allium* species, a voucher specimen was deposited at the Herbarium of Babes-Bolyai University, Cluj-Napoca, Romania (*Allium sativum* L.—CL666161; *Allium fistulosum* L.—CL 669061).

### 3.2. Allium Extract Preparation

Different parts of the plants were used for the extract preparation, leaves for *Allium fistulosum* and bulbs for *A. sativum*. The cold repercolation method at room temperature for 3 days was used for preparing *Allium* plant extract [47] from small plant fragments (0.5–1 cm) and 70% ethanol (Merck, Bucharest, Romania).

The fluid extract (w:v/g:mL) obtained by filtration was 1:1.2 with 30% ethanol for *A. fistulosum* and 1:1 with 20% ethanol for *A. sativum*. Data regarding the chemical composition of *A. sativum* extract was previously published by Pârvu et al. [53].

### 3.3. Phytochemical Characterization of Extracts

The phytochemical characterization was performed according to the method that was previously published by Pârvu et al. [6]. Two HPLC-DAD-MS protocols were optimized to separate and determine the main phytoconstituents as allicin and alliin and polyphenolic compounds [3]. Total phenolic content (TPC) was determined using the Folin Ciocalteu reagent [72] and total tiosulfinate content (TTC) by a kinetic assay previously described [72,73].

### 3.4. Cells and Reagents

The normal human fibroblasts cell line BJ, colorectal cancer cell line DLD-1, squamous cell lung carcinoma cell line SK-MES-1, triple negative breast cancer cell line MDA-MB-231, and double positive breast cancer cell line MCF-7 were purchased from the American Type Culture Collection (ATCC) (Manassas, VA, USA) and the normal human keratinocyte HaCaT was acquired from the Cell Line Service of the German Cancer Research Centre in Heidelberg, Germany. BJ and MCF-7 were maintained in minimum essential medium Eagle—MEM (Sigma Aldrich, ST. Louis, MI, USA), MDA-MB-231, DLD-1, and SK-MES-1 in RPMI (Sigma Aldrich, St. Louis, MI, USA), and HaCaT in Dulbecco’s modified Eagle medium—DMEM with low glucose (Sigma Aldrich, St. Louis, MI, USA) [74,75], all supplemented with 10% fetal bovine serum (Gibco, Grand Island, NY, USA), 1% glutamine (Gibco, Grand Island, NY, USA), and 1% penicillin-streptomycin (Gibco, Grand Island, NY, USA).

### 3.5. MTT Assay on Normal and Tumor Cells

Cytotoxicity study of *Allium* extracts on normal and tumor cell lines were performed using MTT assay (Khazei et al., 2017). Cells were plated at 1 × 10^4^ cells per well in a 96-well plate. After 24 h of incubation, cells were treated with *A. sativum* and *A. fistulosum* extract and incubated for 24 h at 37 °C in a humidified incubator with 5% CO_2_. After 24 h, the cell viability rate was determined using a 3-(4,5-dimethylthiazol-2-yl)-2,5-diphenyltetrazolium bromide (MTT) assay at 570 nm (Sigma-Aldrich, St. Louis, MI, USA). The plates were read with a TECAN SPARK10M (TECAN, Austria GmbH, Grodig, Austria).

The cell viability was calculated by the following equation: % Cytotoxicity = Absorbance of treated cells/Absorbance of negative control × 100. The 50% of the growth inhibition concentration (IC_50_) was calculated from a plotted dose–response curve [36].

### 3.6. Confocal Microscopy—Morphological Analysis

Cells were processed following a triple staining protocol in order to highlight the mitochondrial networks, the cytoskeleton and the nucleus, using Mitotracker-Red, Phalloidin-FITC, and DAPI according to the protocol that was previously published by Budisan et al. [51]. Mitotracker (Thermo Fisher Scientific, Waltham, MS, USA) was added while the cells were still viable, after 90 min of incubation at 37 °C the cells were fixed with 4% paraformaldehyde (Sigma-Aldrich, St. Louis, MI, USA) and the membranes were treated with 0.05% Triton X (Sigma-Aldrich, St. Louis, MI, USA). After Triton X treatment, Phalloidin-FITC (Cytoskeleton, Denver, CO, USA) was added and the samples were incubated for 30 min at room temperature. After Phalloidin incubation, 100 μM DAPI solution (Thermo Fisher Scientific, Waltham, MS, USA) was added for one minute. Between each step the cells were washed three times with phosphate saline buffer 1X (Gibco, Grand Island, NY, USA). Cells were analyzed under an Olympus FLUOVIEW FV1200 laser scanning fluorescence confocal microscope (Olympus, Tokyo, Japan).

### 3.7. Biochemical Determination of LDH, CAT, and Caspase 3

Cells were incubated at 1 × 10^5^ cells per well in a 12-well plate at a final volume of 2000 μL. After 24 h incubation, cells were treated with extracts using three different concentrations (10% extract in the media, the IC_50_ concentration and one concentration below IC_50_). The supernatant was collected after 24 h incubation with extracts. To obtain the cell homogenate, 2000 μL of fresh media was added to each well, then the plates were frozen and thawed three times to dissociate the cell membranes, and the cell extract was collected for further analysis. Lactate-dehydrogenase (LDH) and catalase (CAT) [76], from supernatant samples were analyzed. LDH activity was measured at 340 nm using the LDH Activity Kit for in-vitro use (Biomaxima, Lublin, Poland) and the samples were read with a spectrophotometer (Biochemical Systems International, Model 2000 Evolution, Arezzo, Italy). Catalase activity was measured from the supernatant at 240 nm using an UV-Vis VWR UV-1600PC spectrophotometer (VWR, Radnor, PA, USA). Caspase 3 (Casp3) was determined from the cell supernatant using Human Caspase-3 (Activated) ELISA Kit from Invitrogen (Carlsbad, CA, USA) and the plate was read using a TECAN SPARK10M (TECAN, Austria GmbH, Grodig) according to the assay protocol and the results were calculated using Prism 8 software (San Diego, CA, USA).

### 3.8. Cell Death Investigation by Flow Cytometry

Cells were incubated at 5 × 10^4^ cells per well in a 24-well plate at a final volume of 1500 μL. After 24 h incubation, cells were treated with extracts using three different concentrations (5% extract in the media, the IC_50_ concentration, and one concentration below IC_50_). For detaching cells, Trypsin-EDTA was used, and cells were washed three times with PBS1X, centrifuged and then the cells were resuspended in 200 μL of binding buffer. Annexin V-FITC (ExBIO, Prague, Czech Republic) and propidium iodide (PI) were added according to the manufacturer’s protocol, 5 μL Annexin V and 5 μL of propidium iodide then 15 min of incubation at room temperature was performed to allow staining.

After incubation, cells were centrifuged and resuspended in 100 μL of 1X annexin binding buffer and the samples were immediately read using a FACS BD Flow cytometer (BD, San Jose, CA, USA) and the results were calculated with Prism8 software.

### 3.9. Statistical Analysis

Viability rate was calculated using Prism 8 software, two-way ANOVA, and Bonferroni’s post-hoc test was considered statistically significant at *p* < 0.05 and was interpreted as follows: * *p* < 0.05, ** *p* < 0.01, and *** *p* < 0.001 when comparisons were made with the C group. All the normal cells were compared to an untreated group as the control and the results were expressed as the mean value ± SD. The results were expressed as * *p* < 0.05, ** *p* < 0.01, and *** *p* < 0.001 with Student’s t-test by comparison with the control group (untreated) for the MTT assay on tumoral cells (GraphPad Software, San Diego, CA USA).

## 4. Conclusions

*Allium sativum* and *A. fistulosum* inhibited human fibroblasts and human keratinocytes growth when a high concentration of these extracts was applied. The MTT assay indicated a dose dependent growth inhibition for both extracts. Further observations of morphological changes in BJ and HaCaT cells, using a triple staining protocol, revealed the mitochondrial network, cytoskeleton, and nuclear modification, also the cell population was reduced in a dose dependent manner. Morphological modifications are indicators of the cell death induced by garlic and Welsh onion extracts at high doses of the extract. The flow cytometry analysis confirmed that cell death was induced at high extract concentrations, mostly by necrosis, correlating the results with the Casp3, LDH, and CAT analysis.

The extracts had different allicin concentrations, showing an antiproliferative effect against four tumor cell lines, with different IC_50_ values, depending on the concentration of allicin according to the phytochemical analysis of both extracts. We believe that the use of isolated bioactive sulfuric compounds, such as allicin, therefore can be used against tumor cells with promising results. We aim to develop further in vivo studies focused on the molecular interaction of allicin and other bioactive compounds with key molecules that control biological processes in the cells to determine the therapeutical potential of *Allium* bioactive compounds

## Figures and Tables

**Figure 1 molecules-26-00574-f001:**
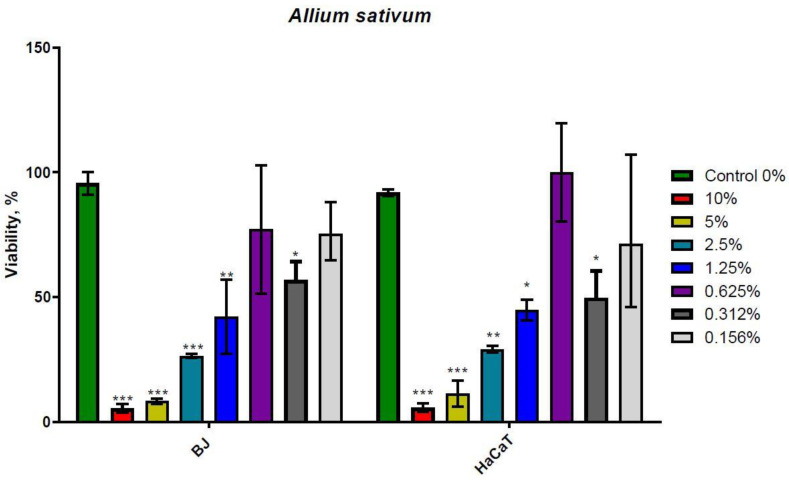
Twenty-four hour MTT Assay on BJ and HaCaT treated with *Allium sativum*. * *p* < 0.05, ** *p* < 0.01, and *** *p* < 0.001 when comparisons were made with the C group. Garlic hydroalcoholic extract induces cell death in a dose dependent manner in both cell lines. The control group is represented by the untreated cells, which were cultivated with standard cell culture medium.

**Figure 2 molecules-26-00574-f002:**
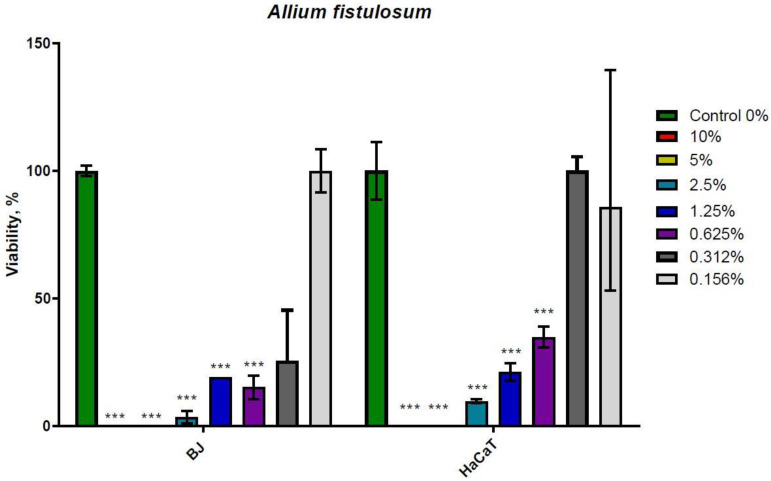
Twenty-four hour MTT Assay on BJ and HaCaT treated with *Allium fistulosum*. *** *p* < 0.001 when comparisons were made with the C group. Welsh onion hydroalcoholic extract induces cell death in a dose dependent manner in both cell lines. The control group is represented by untreated cell, which were cultivated with standard cell culture medium.

**Figure 3 molecules-26-00574-f003:**
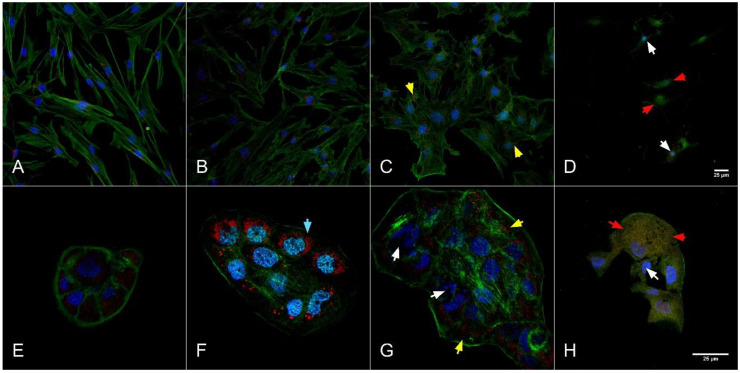
*Allium sativum* cytotoxic effect on BJ cells with a 40× objective ((**A**)—Control group, (**B**)—0.5% extract, (**C**)— 50% of the growth inhibition concentration (IC_50_) dose, and (**D**)—10% extract) and HaCaT cells with 100× objective ((**E**)—Control group, (**F**)—0.1% extract, (**G**)—IC_50_ dose, and (**H**)—10% extract). Cells were labeled with different strainers, DAPI for the nucleus (blue), Mitotracker for the mitochondria (red), and Phalloidin-FITC for the cytoskeleton (green). Nuclear damage—White arrows; Cytoskeleton disruption—Red arrows; Mitochondrial network reduced intensity—Yellow arrows; Mitochondrial intensified activity—Turquoise arrows.

**Figure 4 molecules-26-00574-f004:**
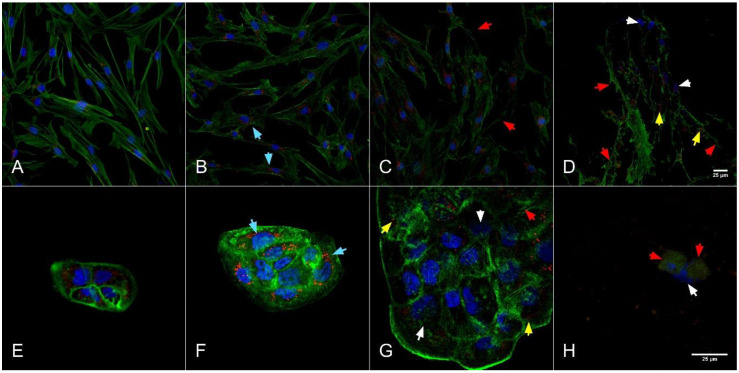
*Allium fistulosum* cytotoxic effect on BJ cells with 40× objective ((**A**)—Control group, (**B**)—0.5% extract, (**C**)—IC_50_ dose, and (**D**)—10% extract) and HaCaT cells with 100× objective ((**E**)—control group, (**F**)—0.1% extract, (**G**)—IC_50_ dose, and (**H**)—10% extract). Cells were labeled with different strainers, DAPI for the nucleus (blue), Mitotracker for the mitochondria (red), and Phalloidin-FITC for cytoskeleton (green). Nuclear damage—White arrows; Cytoskeleton disruption—Red arrows; Mitochondrial network reduced intensity—Yellow arrows; Mitochondrial intensified activity—Turquoise arrows.

**Figure 5 molecules-26-00574-f005:**
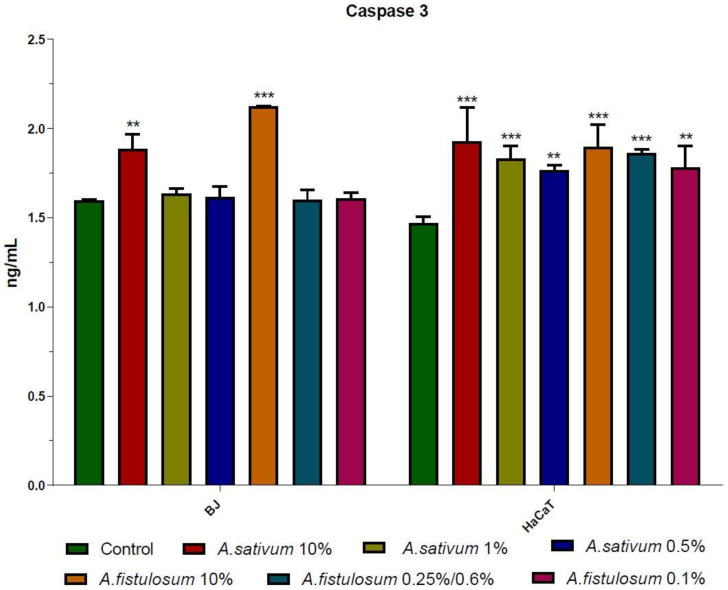
For BJ cells, increased Casp3 was observed for 10% *Allium sativum* extract (100 mg/mL) and for 10% *A. fistulosum* extract (83 mg/mL), while for other concentrations there was no difference compared to the control group. As for HaCaT cells Casp3 was activated in a dose-dependent manner compared with the control group. Even if the statistical significance was very high, the numeric value of Casp3 activity was not so high and the results must be correlated with other parameters analyzed. ** *p* < 0.01, and *** *p* < 0.001 when comparisons were made with the C group.

**Figure 6 molecules-26-00574-f006:**
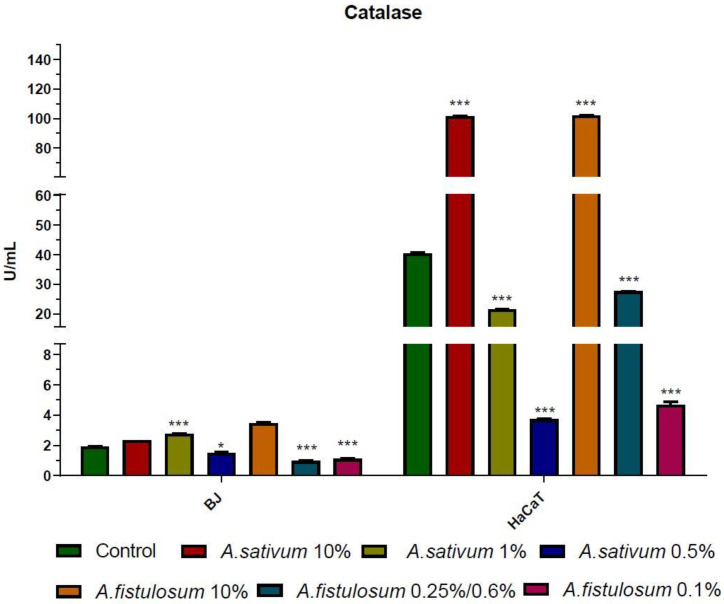
Catalase was increased around the IC_50_ value of *Allium sativum* extract in BJ cells (1% extract equivalent to 10 mg/mL) and decreased in treatment with *A. fistulosum*; the numeric value of CAT activity is not so high, compared to HaCaT cells. Keratinocytes are more sensitive to *A. sativum* and *A. fistulosum* extract and the effect is different for each dose. * *p* < 0.05 and *** *p* < 0.001 when comparisons were made with the C group.

**Figure 7 molecules-26-00574-f007:**
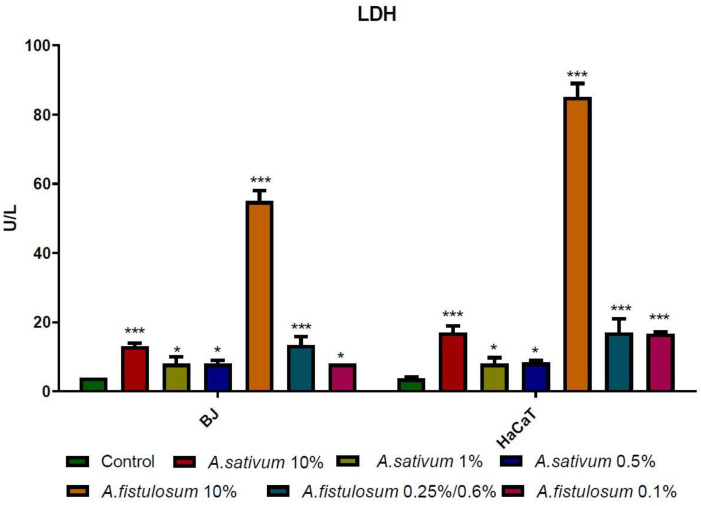
For both cell lines, *Allium sativum* and *A. fistulosum* extracts enhanced LDH activity in a dose dependent manner, with the most intensified activity for the 10% of extract compared with the control group. *A. fistulosum* increased LDH activity more than garlic extract in both cell lines, indicating that the chemical composition of the extracts is a very important player in how different molecules are modulated. * *p* < 0.05 and *** *p* < 0.001 when comparisons were made with the C group.

**Figure 8 molecules-26-00574-f008:**
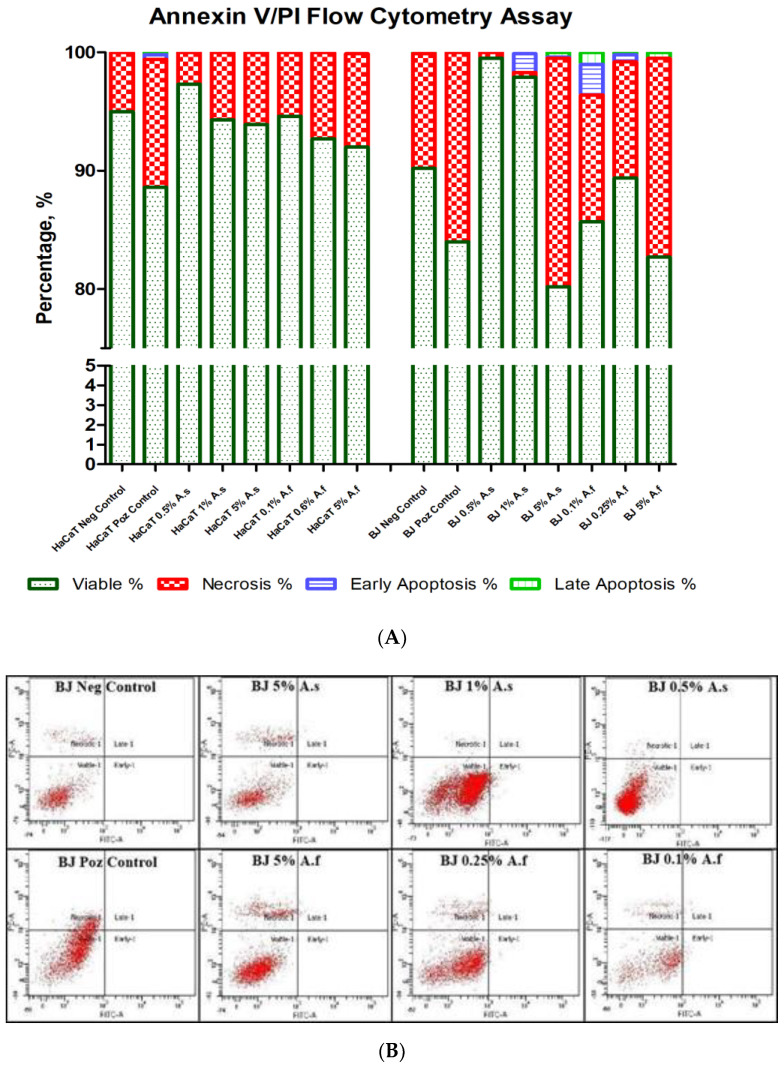
Annexin V/PI Flow Cytometry Assay for BJ and HaCaT cell lines treated with different concentrations of *Allium sativum* and *A. fistulosum* extracts. Annexin V and PI were added to each sample. Negative controls were not treated with extracts and positive controls were supplemented with DMSO at a final volume of 10% in growth media. All extract concentrations were selected to be similar to other concentrations that were used for other determinations. Both cell lines were incubated for 24 h with certain *Allium* extracts. (**A**)—graph representing population percentages; (**B**)—BJ scatter plot; (**C**)—HaCaT scatter plot. A.s—*Allium sativum*; A.f.—*Allium fistulosum*; Neg Control—negative control; Poz Control—positive control.

**Figure 9 molecules-26-00574-f009:**
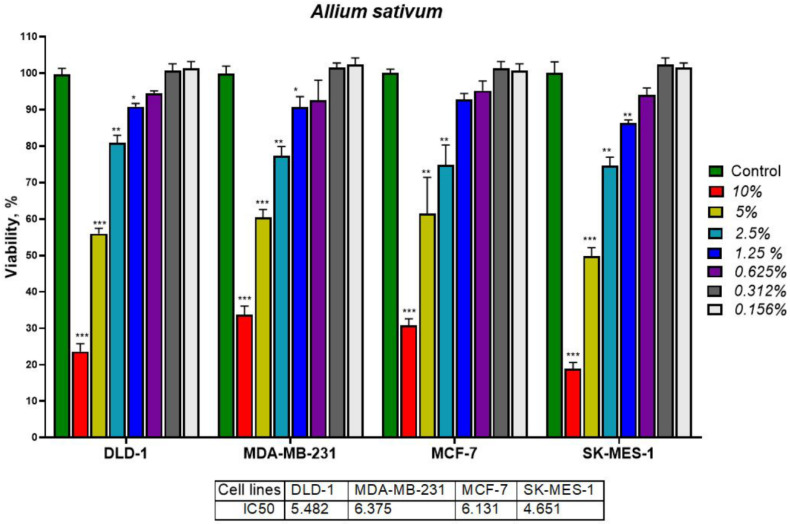
MTT assay for DLD-1, MDA-MB-231, MCF-7, and SK-MES-1 cell lines treated with different concentrations of *Allium sativum* extract. The results were expressed as * *p* < 0.05, ** *p* < 0.01, and *** *p* < 0.001 with Student’s *t*-test by comparison with control group (untreated). The samples treated with 10% extract in the culture medium (equivalent to 100 mg mL^−1^) showed highly significant inhibition, with a significant inhibitory effect even at 1% extract. SK-MES-1 was the most sensitive cell line with 4.651% extract as the IC_50_ value (equivalent to 46 mg/mL) and MDA-MB-231 the triple negative breast cancer cell line being the less sensitive cell line with 6.375% extract (equivalent to 63 mg/mL) as the IC_50_ value.

**Figure 10 molecules-26-00574-f010:**
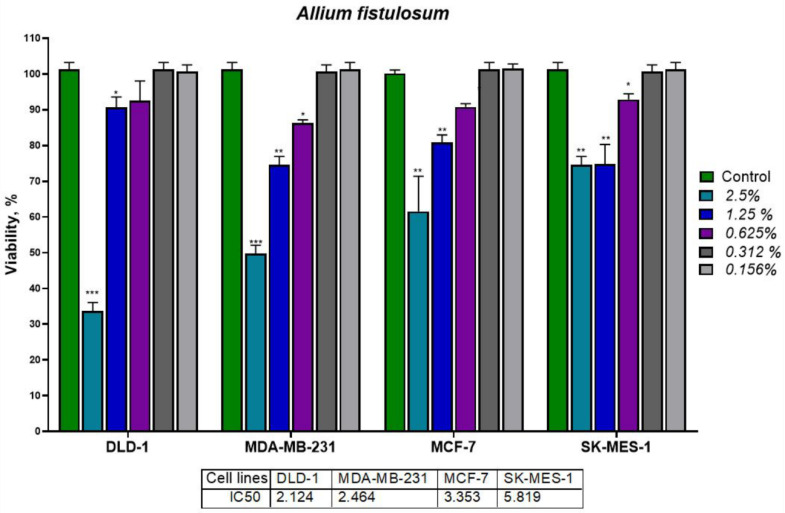
MTT assay for DLD-1, MDA-MB-231, MCF-7, and SK-MES-1 cell lines treated with different concentrations of *Allium fistulosum* extract. The results were expressed as * *p* < 0.05, ** *p* < 0.01, and *** *p* < 0.001 with Student’s *t*-test by comparison with the control group (untreated). Due to the pH changes induced by the 10% and 5% concentrations in the cell culture medium observed by the phenol red color change, the MTT assay was performed between 2.5% and 0.156% extract. The IC_50_ values were between 5.819% extract in cell culture medium for SK-MES-1 and 2.123% extract in cell culture medium for DLD-1 cell line.

**Table 1 molecules-26-00574-t001:** Elution time, analytical method characteristics, and determined concentrations of alliin, allicin, and phenolic acids and flavonoids in the *Allium* extract samples (results expressed as the mean value of 4 analyzed samples, *n* = 4).

No	Compounds	t_elution_ (min)	R^2^	LOD (µg/mL)	*A. sativum* (µg/mL) *	*A. fistulosum* (µg/mL)
1	Alliin	3.77	0.9999	5.8	1410 ± 50	145 ± 15
2	Allicin	15.40	0.9999	14.1	380 ± 15	20 ± 5
3	Gentisic ac.	8.13	0.9997	3.4	60 ± 5	-
4	Chlorogenic ac.	9.15	0.9995	4.6	65 ± 5	-
5	4-hydroxybenzoic	10.31	0.9999	2.3	25 ± 3	-
6	Rutin	11.39	0.9998	2.7	-	215 ± 3
7	Isoquercitrin	11.91	0.9999	2.2	-	280 ± 3
8	*p*-Coumaric	12.38	0.9999	1.9	44 ± 4	-
9	Quercitrin	12.72	0.9998	2.7	-	95 ± 3
10	Ferulic ac.	12.85	0.9999	2.0	-	230 ± 2
11	Quercetin	16.35	0.9997	3.3	-	26 ± 3
12	Kaempferol	19.77	0.9998	2.7	-	30 ± 3

LOD—limit of detection, R^2^—coefficient of determination for the calibration curves (at six levels of concentrations). Indicated intervals represent the average ± standard deviations (*n* = 4). * The information regarding *Allium sativum* was previously published by Pârvu et al. [10].

## Data Availability

Not applicable.

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
