# Peer review of "Phytochemical Analysis and In Vitro Effects of Allium fistulosum L. and Allium sativum L. Extracts on Human Normal and Tumor Cell Lines: A Comparative Study"

_molecules, 2021, doi:10.3390/molecules26030574_

Round 1
Reviewer 1 Report
Molecules-1073960
A Brief Summary
The work is very suitable to publish in Molecules. The authors tried to establish that the extracts of Allium fistulosum L. and Allium sativum L. are potentially important for the antitumour or anticancer effects. Some spelling mistakes have been found entire the MS. They should consider some minor points for upgrading their manuscript.
Minor revision
Abstract
The IC50 value of each cell line must include in the abstract as results
Introduction:
The introduction is very long. Please present the rationale of the study with the appropriate discussion which is highly relevant to your work. For example, the paragraph seems to be exaggeration '''Medical plants .........conventional therapy.''
Is it medical plants
Check the spell ‘’per se or combined’’
Results
Figure 1 and 2: The control bar legends have been found no footnote.
In the cytotoxic analysis, which compounds used as a positive control against the cell cytotoxicity assay for comparing the activity.
The authors mention only garlic or onion extract but not specifically mention the which type of extract is, is it 70% ethanol or others.
Discussion
The phytochemical effect against the cytotoxicity has not been discussed.
Conclusion:
The future perspective, applications or direction of the study has not been discussed. Also, the impact of the phytochemical on the cytotoxicity did not portrait.

Author Response
Dear reviewer,
Thank you for taking your time to review our paper, for your comments and suggestions which strongly improved the manuscript. We will answer now, point-by-point after each remark, also we added with Track Changes all your observations with our response in the main text.
Comments and Suggestions for Authors
Molecules-1073960
A Brief Summary
The work is very suitable to publish in Molecules. The authors tried to establish that the extracts of Allium fistulosum L. and Allium sativum L. are potentially important for the antitumour or anticancer effects. Some spelling mistakes have been found entire the MS. They should consider some minor points for upgrading their manuscript.
Minor revision
Abstract
The IC50 value of each cell line must include in the abstract as results
R: Thank you for your observation, the IC50 values were added in the abstract. Indeed, the abstract information is now clear and will help the readers to understand the study.
Introduction:
The introduction is very long. Please present the rationale of the study with the appropriate discussion which is highly relevant to your work. For example, the paragraph seems to be exaggeration '''Medical plants .........conventional therapy.''
R: Thank you for your comment. Indeed, the paragraph regarding medical plants is a bit exaggerated. After a close check, we decided that we can delete this paragraph and keep the paragraph regarding Natural compounds which also summarizing the paragraph that will be deleted.
As regarding the Introduction chapter, we would like to keep the other paragraphs, as we limited this chapter to one page, summarizing more than 45 sources.
“Natural compounds represent potential candidates in fighting against various diseases. A pure compound, isolated and well characterized, can serve as a therapeutic agent alone or in combination with conventional therapy, due to their biological properties. Medical plants are a rich source of nutrients and bioactive compounds, most of them serving as models for synthesized analogues or used after extraction for investigating their different biological properties on different pathologies [45-47]. During the last decades, natural compounds were investigated in different studies, most of them involving their antiproliferative effects or their ability to inhibit tumor features [48-51].”
Check the spell ‘’per se or combined’’
R: Thank you for this remark. We changed “per se” with “individual as single agent treatment”.
Results
Figure 1 and 2: The control bar legends have been found no footnote.
R: Thank you for your observation. We mentioned in Figure legend that the Control group represents untreated cells. “The control group is represented by the untreated cells, which were cultivated with standard cell culture medium”.
In the cytotoxic analysis, which compounds used as a positive control against the cell cytotoxicity assay for comparing the activity.
R: For the cytotoxic analysis we did not use a certain compound as a positive control (e.g. MTT assay) the evaluation was performed compared to the untreated group, while when investigating the viable cells vs dead cells by Flow Cytometry, we used DMSO as a positive control to verify the necrotic cells, due to the necrosis features that were visible on microscopy analysis.
The authors mention only garlic or onion extract but not specifically mention the which type of extract is, is it 70% ethanol or others.
R: Thank you for your observation. We added in the text that garlic and Welsh onion extracts are hydroalcoholic. As mentioned in the Materials and Methods section, Allium sativum extract stock solution has 30% Ethanol, and Allium fistulosum extract stock solution has 20% Ethanol.
Discussion
The phytochemical effect against the cytotoxicity has not been discussed.
R: Thank you for your remark. We modified the results and discussion section, by merging them, and regarding your observations, we discussed on extract chemical composition and the literature that reported the biological effects of allicin and polyphenols from Allium species. Please find the paragraphs below:
“The dose-dependent inhibitory effect of A. sativum and A. fistulosum extracts is related to the concentration of allicin. Gruhlke et al., evaluated the inhibitory effect of allicin in garlic juice and synthesized allicin on lung adenocarcinoma (A549 cell line), human mammary carcinoma (MCF-7 cell line), human colorectal carcinoma (HT29 cell line), mouse fibroblasts (NIH 3T3 cell line) and human umbilical vein endothelial cells (HUVEC) and the results indicate an inhibitory rate on all cell lines, in a dose-dependent manner [20]. Furthermore, Allium extracts are rich in polyphenols which play important role in the biological effects of garlic and Welsh onion. The antitumor effect of both Allium extracts is mainly attributed to allicin, while polyphenols are enhancing this biological effect. Fujimoto et al., demonstrated that polyphenols have cytotoxic effect in a dose-dependent manner both on normal and tumor cells, and at a well-established concentration, polyphenols could inhibit tumor cell without affecting normal cells [71].
Polyphenols, in a dose-dependent manner, can induce cell death by inducing apoptosis [71]. Sulfuric compounds, like allicin, have a different effect on cell signaling pathways. Allicin inhibits cell growth and induces apoptosis via the ERK-dependent pathway [31] or can activate both intrinsic and extrinsic apoptotic pathways [25,33,70]. Moreover, allicin can inhibit tumor cell progression by suppressing the HIF pathway [66]. Allicin suppresses cervical cancer cells invasion and migration via inhibiting NRF2 [68]. Furthermore, allicin has an immunostimulatory effect via Colec12, MARCO and SCARB1 receptors [19]. A similar proapoptotic effect was observed in the case of diallyl trisulfides, which are part of Allium plats chemical composition, Shin et al., demonstrated that diallyl trisulfides induce apoptosis in caspase-dependent manner and create cross-talks with PI3K/Akt and JNK pathways [58].
Both Allium extracts that were investigated in our study are rich in allicin and polyphenols and showed an inhibitory effect in a dose-dependent manner both on normal and tumor cells. At high concentrations, above the concentration obtained for the IC50, cell damage was induced and the LDH and CAT activity was increased. On the other hand, at lower concentrations the cellular stress was reduced, LDH and CAT activity were significantly decreased compared to the 10% extract. The inhibitory effect is related to the chemical composition of the extracts. The cytotoxicity could be reduced by choosing a therapeutic dose that is not toxic to normal cells and by isolating or synthesizing bioactive compounds from the extracts, compounds that are responsible for the biological effect of these plants.”
Conclusion:
The future perspective, applications or direction of the study has not been discussed. Also, the impact of the phytochemical on the cytotoxicity did not portrait.
R: Dear reviewer, we modified the discussion section and discussed about the cytotoxicity that is induced in a dose-dependent manner. We cited papers that have reported similar effect on allicin and polyphenols, on normal or tumor cells, moreover, all the papers have mentioned that is very important to select a proper therapeutic dose that cannot induce normal cell damage, mostly by using isolated bioactive compounds or by reducing the extract concentration.
We wish to continue our studies on in vivo models to determine the interaction between cellular key molecules from different pathways and the bioactive compounds from Allium plants.

Reviewer 2 Report
The manuscript “Phytochemical analysis and in vitro effects of Allium fistulosum L. and Allium sativum L. extracts on human normal and tumor cell lines: a comparative study” by Bogdan Țigu et al. investigated the effect of two extracts from allium species on cell culture lines (tumor and normal).
The manuscript focuses on the cytotoxicity and the mechanisms governing the molecular effects. The authors did a great job, but there are minor changes to be done to improve their manuscript’s final scientific quality.
- In the results section, avoid using references and discuss the reults. This should be done in the discussion section. The authors can merge bot sections if they prefer it.
- The style of the graphs could be change to show higher uniformity.
- Why authors studied the mechanisms only for normal cells? What about the tumor cells?
- The discussion should be comprehensively revised. The authors could go deep into the relationship between the phytochemicals in the extracts and the observed effects.
- If the extracts are cytotoxic in normal cell lines, how can they be used in tumor cells without generating adverse effects?
- The authors should perform experiments to complete the mechanisms inducing cytotoxic effects in tumor cell lines.
Hence, this manuscript is not ready to be published in Molecules by MDPI.
Author Response
Dear reviewer,
Thank you for taking your time to review our paper, for your comments and suggestions which strongly improved the manuscript. We will answer now, point-by-point after each remark, also we added with Track Changes all your observations with our response in the main text.
Comments and Suggestions for Authors
The manuscript “Phytochemical analysis and in vitro effects of Allium fistulosum L. and Allium sativum L. extracts on human normal and tumor cell lines: a comparative study” by Bogdan Țigu et al. investigated the effect of two extracts from allium species on cell culture lines (tumor and normal).
The manuscript focuses on the cytotoxicity and the mechanisms governing the molecular effects. The authors did a great job, but there are minor changes to be done to improve their manuscript’s final scientific quality.
- In the results section, avoid using references and discuss the reults. This should be done in the discussion section. The authors can merge bot sections if they prefer it.
R: Thank you for your observation. We merged the results and discussion sections. Also we discussed regarding the pathways in which the bioactive compounds from Allium extracts are involved, by citing the literature.
- The style of the graphs could be change to show higher uniformity.
R: Thank you for your comment. Indeed, a higher uniformity of graphs is necessary. We changed figure 1,2,5,6 and 7 in similar style as figure 9 and 10. However, we would like to keep figure 8 with flow cytometry data in its current form.
- Why authors studied the mechanisms only for normal cells? What about the tumor cells?
R: Regarding this aspect we would like to answer related to another study that was performed by our research team. In the presented manuscript we evaluated the effect of both Allium extracts on normal cells (BJ and HaCaT) investigating the inhibitory effect on cell proliferation, the morphological changes that were induced by the treatment and we wanted to show how the extracts can reduce cell population by inducing oxidative stress (by evaluating CAT activity) or by inducing cell death (by evaluating the membrane integrity biomarker LDH and the apoptosis effector Caspase 3). Moreover, we showed the ratio between viable and dead cells by flow cytometry. These results made us curious and we started to evaluate the effect of extracts on tumor cells, as presented in figure 9 and 10. The results indicated that the extract is reducing proliferation rate in a dose dependent manner in tumor cells, similar to normal cells.
As we presented in the paper, we used various concentrations of extract, extracts which have chemical composition rich in nutrients and compounds that were reported in literature with antitumor properties (sulfuric compounds and polyphenols). Due to the high concentration of Allicin in A. sativum extract (and lower concentration in A. fistulosum) we decided that is better to synthesize Allicin and to use it as antitumor agent, therefore we started a new experiment which is completing the herein presented study.
The study regarding Allicin as antitumor agent was published in 2020, in Molecules (doi: 10.3390/molecules25081947) and we demonstrated the synergistic effect of Allicin and 5-FU on DLD-1 and SK-MES-1 cell lines, two of the cell lines that were evaluated in the submitted manuscript.
We now seek to develop a study protocol to further investigate Allicin, 5-FU and other antitumor agents on in vivo models that will develop tumors after inoculating DLD-1 and SK-MES-1 cells. We think that the results on in vivo models using Allicin, not the Allium extracts, will provide significant data regarding Allium bioactive compound (allicin) antitumor properties.
Thank you for asking regarding the tumor cells. We do hope that our answer is complete.
- The discussion should be comprehensively revised. The authors could go deep into the relationship between the phytochemicals in the extracts and the observed effects.
R: Thank you for your remark. We discussed regarding the molecular implications of allicin and polyphenols in biological processes. Please find the changes in the main text and below:
“The dose-dependent inhibitory effect of A. sativum and A. fistulosum extracts is related to the concentration of allicin. Gruhlke et al., evaluated the inhibitory effect of allicin in garlic juice and synthesized allicin on lung adenocarcinoma (A549 cell line), human mammary carcinoma (MCF-7 cell line), human colorectal carcinoma (HT29 cell line), mouse fibroblasts (NIH 3T3 cell line) and human umbilical vein endothelial cells (HUVEC) and the results indicate an inhibitory rate on all cell lines, in a dose-dependent manner [20]. Furthermore, Allium extracts are rich in polyphenols which play important role in the biological effects of garlic and Welsh onion. The antitumor effect of both Allium extracts is mainly attributed to allicin, while polyphenols are enhancing this biological effect. Fujimoto et al., demonstrated that polyphenols have cytotoxic effect in a dose-dependent manner both on normal and tumor cells, and at a well-established concentration, polyphenols could inhibit tumor cell without affecting normal cells [71].
Polyphenols, in a dose-dependent manner, can induce cell death by inducing apoptosis [71]. Sulfuric compounds, like allicin, have a different effect on cell signaling pathways. Allicin inhibits cell growth and induces apoptosis via the ERK-dependent pathway [31] or can activate both intrinsic and extrinsic apoptotic pathways [25,33,70]. Moreover, allicin can inhibit tumor cell progression by suppressing the HIF pathway [66]. Allicin suppresses cervical cancer cells invasion and migration via inhibiting NRF2 [68]. Furthermore, allicin has an immunostimulatory effect via Colec12, MARCO and SCARB1 receptors [19]. A similar proapoptotic effect was observed in the case of diallyl trisulfides, which are part of Allium plats chemical composition, Shin et al., demonstrated that diallyl trisulfides induce apoptosis in caspase-dependent manner and create cross-talks with PI3K/Akt and JNK pathways [58].
Both Allium extracts that were investigated in our study are rich in allicin and polyphenols and showed an inhibitory effect in a dose-dependent manner both on normal and tumor cells. At high concentrations, above the concentration obtained for the IC50, cell damage was induced and the LDH and CAT activity was increased. On the other hand, at lower concentrations the cellular stress was reduced, LDH and CAT activity were significantly decreased compared to the 10% extract. The inhibitory effect is related to the chemical composition of the extracts. The cytotoxicity could be reduced by choosing a therapeutic dose that is not toxic to normal cells and by isolating or synthesizing bioactive compounds from the extracts, compounds that are responsible for the biological effect of these plants.”
- If the extracts are cytotoxic in normal cell lines, how can they be used in tumor cells without generating adverse effects?
R: Thank you for this observation. The extract showed a dose-dependent cytotoxicity. Therefore, a lower concentration of these extracts should be used as treatment. For antibacterial, antifungal or antioxidant properties of Allium extracts, lower concentrations can be used. As we wanted to further focus on the antitumor properties of these Allium extracts, we observed that at concentrations that are not cytotoxic on normal cells, the antitumor effect is not satisfying, tumor cells had higher proliferation rate compared to normal cells, which is influencing the experiment, more cells are dividing than cells that are undergoing cell death after treatment. We had two choices, to try different extracts, or to evaluate their bioactive compounds. We decided, as we answered before, to synthesize allicin and to use it as antitumor agent. The experiments that are presented in this manuscript are part of a large study which includes also the previous published paper (doi: 10.3390/molecules25081947).
- The authors should perform experiments to complete the mechanisms inducing cytotoxic effects in tumor cell lines.
R: We understand that other experiments can increase the scientific value of the presented study. As we answered to your previous questions/comments, the information presented in this paper is part of a larger study, after obtaining these results, we understood that the bioactive compounds that are present in Allium plants can be used, after synthesis or purification, as antitumor agents. The extracts are rich in many other compounds and we believe that is better to use the compound not the whole extract for the antitumor experiment. We believe that allicin has the potential to be used as adjuvant therapeutic or after synthesis and stabilization as a single agent treatment against cancer cells. In the last decades, many natural compounds were isolated after studying the extracts of the plants and some of them are currently used as therapeutics in cancer treatment (like Paclitaxel and Vinorelbine).
Hence, this manuscript is not ready to be published in Molecules by MDPI.

Reviewer 3 Report
Authors should pay attention on significant number of technical errors through a whole Manuscript, including references list.
Also, I would like to ask authors to activate Line numbers option in revised version. In current version it was quite hard to make a comments. So, I was forced to include comments in pdf file of Manuscript which is attached.

Author Response
Dear reviewer,
Thank you for taking your time to review our paper, for your comments and suggestions which strongly improved the manuscript. Your suggestion had significantly improved our manuscript and we are grateful that we were able to apply them in this text.
We will answer now, point-by-point after each remark, also we added with Track Changes all your observations with our response in the main text.
We added the line numbers in the manuscript.
Comments and Suggestions for Authors
Authors should pay attention on significant number of technical errors through a whole Manuscript, including references list.
Also, I would like to ask authors to activate Line numbers option in revised version. In current version it was quite hard to make a comments. So, I was forced to include comments in pdf file of Manuscript which is attached.
R: Thank you for your observations. We will answer point-by point according to all your observations.
Response to reviewer 3:
Abstract:
Precise which part of plants was used for analysis.
R: We changed the information. We used garlic bulbs and Welsh onion leaves for preparing the extract. The information is now available in the abstract.
Put italic to p-coumaric.
R: we added italic letter to this compound in abstract and elsewhere this compound was mentioned in the text.
Regarding high concentration - I think it should be concentration in singular? Or include some other concentration in brackets.
R: concentrations were added (10%, 5%, 2.5%, 1.25%)
Regarding high concentration - I think it should be concentration in singular? Or include some other concentration in brackets
R: concentrations were added (5%, 2.5%, 1.25%)
Introduction:
All onion to A. cepa
R: onion was added.
Regarding the word Alliums - I am not sure that Latin name can be given as English plural with s at the end?
R: - The word was changed accordingly. We added “Allium species”.
Regarding word Interestst – check word
R: we changed it into “interest”
Results:
Are the results given in Table 1 mean value for these 4 extracts or what? It is not clear in current version. Please clarify this issue.
R: We changed the text, we hope that the use of the mean value between 4 analyzed samples could be visible. “Results expressed as the mean value of 4 analyzed samples”
Also below table 1 we explained the intervals represented as average+/- SD where n=4 samples.
I suppose that you are here talking about 10% extract? But it must be specified since on Figure 1 there are also other values.
R: Thank you for your remark. Hoping that I fully understood your comment, we have presented in figure 1 and figure 2 seven serial diluted samples, which represent the experimental groups for MTT Assay, while in the text we mentioned that the antiproliferative effect was expressed in a dose-dependent manner and we wrote the IC50 value which is 0.8841% Garlic extract for BJ cells and 1.086% for HaCaT. Also, we kept the mg/mL unit for equivalent dose calculated based on the extract stock solution concentration presented in Materials and Methods.
And where is the mentioned Table 2? It is not given in Manuscript. Please check and clarify
R: Thank you for these remarks. We managed to change all the words that were indicated. Regarding table 2 mentioned in the text, that was a typing error, there is no table 2 in the text, we changed it to Table 1. Capital letters were also added to all the words that were highlighted in the text.
Authors should explain how they convert these values from % to mg/mL. Especially since there are difference here compared to results from Figure 2.
R: Thank you for your remark. When the extracts were prepared, we used 1 g/mL Garlic Bulbs and 1 g/ 1.2 mL Welsh onion leaves, therefore when converting percentages of extract in cell culture medium we used these concentrations to determine the amount of extract in mg/mL for each concentration obtained after the MTT Analysis.
We added in the text at chapter 2.2
“When calculating the equivalent of extract in mg/mL we used the concentrations of stock solutions for each extract, 1 g garlic bulbs in each mL and 1 g of Welsh onion leaves in each 1.2 mL.”
Superscript is given in incorrect way. In addition the rest of concentration are given as mg/mL so I think this one should be also expressed as mg/mL instead of given version.
R: Thank you for your comment. We changed it in mg/mL.
Split unit from numerical value.
R: Thank you for your remark. We changed it accordingly.
% as unit should not be separate from numerical values. Please apply to all in text.
R: Thank you for your observation. We solved this issue and applied it to all in text.
It can not be over 1.25% since 5% and 10% are over. It can only be equal or below or something like that?
R: Thank you for your observation. We changed the sentence and added “below “and deleted “over”.
Specify for what concentration. You just said above that at IC50 there are no cytotoxic effect for cells.
R: Thank you for your comment. The text was modified and we explained the effects highlighted in Figure 3. Also, below with red text, we added the changes that were made in the main text.
“These images confirm the cytotoxic effect for both extracts at high concentration (10% extract) and less cytotoxic at IC50 concentrations (0.8841% for BJ and 1.086% for HaCaT) and below IC50 (0.5% for BJ and 0.1% for HaCaT), morphological modification with the disrupted cytoskeleton and mitochondrial network is indicating that cell death was induced by necrosis at 10% extract. Morphological changes are less intense at IC50 concentrations and below compared to 10% extract.”
Why this IC value is deferent from the one given above? Please check/clarify/explain.
R: We modified the text between brackets to be clear for the readers which concentrations are at IC50 values.
At the IC50 concentration (10 mg/mL 0.8841% for BJ and 1.086% for HaCaT)
At concentrations below IC50 (0.5% for BJ and 0.1% for HaCaT 5 mg/mL)
But you just said above that IC value had no cytotoxic effect. Please check/correct/explain/clarify.
R: Thank you for your observation, the paragraph was changed. We hope that now well explained that the cytotoxic effect is in a dose-dependent manner, and when lowering the concentration of the extract added to the cells the cell damage is decreased. Also, below with red text we added the changes that were made in the main text.
“These images confirm are indicating a the cytotoxic effect for both extracts when concentration is high (10% extract) and lower cytotoxic effect when the extract concentration is reduced, morphological modification with the disrupted cytoskeleton and mitochondrial network is indicating that cell death was induced by necrosis in the group treated with 10% extracts, while at lower concentrations of extracts cell populations were less affected and the structural damage was highlighted in a dose-dependent manner”
Here previous sentence should be finished and start the new one. In current form it does not make sense with the following part.
R: Thank you for your comment. Sentence was changed (At section 2.6.)
Is it possible to be more specific and to precisely define IC value like you did for previous assays?
R: We added some information regarding the IC50 values, highlighting the most sensitive and the less sensitive cell lines to each extract. Probably this way the information can be better analyzed by readers. Please find below the text that was changed.
The IC50 for both Allium extracts were higher than 1 mg/mL than 1.25% extract (equivalent to more than 1 mg/mL). A. sativum extract had IC50 values of 4.651 % for SK-MES-1 which is the most sensitive cell line to garlic extract, 5.482 % for DLD-1, 6.131 % for MCF-7 and 6.375 % for MDA-MB-231 which is the less sensitive cell line to garlic extract. Welsh onion extract has IC50 values of 2.124% for DLD-1 which is the most sensitive cell line to Welsh onion extract, 2.464% for MDA-MB-231, 3.353% for MCF-7 and 5.819% for SK-MES-1 which is the less sensitive cell line to Welsh onion.
Materials and Methods
Chapter 4.5. - has been instead of was been.
R: Thank you for your remark, we solved this issue.
Chapter 4.6. No green color, use black.
R: Thank you, we solved this issue.
Conclusions
At the first paragraph: Maybe applied is better choice?
R: Thank you, we changed the word “used” with “applied” as suggested.
References:
I think there are repeating errors through a whole list:
- I think that all names of journals should be given in abbreviated forms. Please check MDPI rules.
- Use . after abbreviated words.
- All words in journal's names which are not and, the, etc. should be given with capital letters. For instance, Science, Letters, etc. Please check all.
R: Thank you very much for your remarks regarding the reference list. We solved these issues and all Journal abbreviations were added according to your suggestions.

Round 2
Reviewer 2 Report
The authors have properly answered reviewers' questions and improved their manuscript according to reviwers' suggestions.
The manuscript can be published now.
Reviewer 3 Report
No further comments.